# High-Dose Vitamin C Supplementation as a Legitimate Anti-SARS-CoV-2 Prophylaxis in Healthy Subjects—Yes or No?

**DOI:** 10.3390/nu14050979

**Published:** 2022-02-25

**Authors:** Beata M. Gruber-Bzura

**Affiliations:** Department of Biochemistry and Biopharmaceuticals, National Medicines Institute, 00-725 Warsaw, Poland; b.gruber@nil.gov; Tel.: +48-22-841-39-91

**Keywords:** vitamin C, immune function, SARS-CoV-2, prophylaxis

## Abstract

Vitamin C has a number of acitvities that could contribute to its immune-modulating effects. The only question is whether we should provide ourselves with only the right level of it, or do we need much more during a pandemic? The possibility of reducing the incidence of viral diseases in a well-nourished population through the use of dietary supplements based on vitamin C is not supported in the literature. Despite this, the belief that an extra intake of vitamin C can increase the efficacy of the immune system is still popular and vitamin C is advertised as a remedy to prevent infectious disease. This article refers to the justification of the use of vitamin C in high doses as an anti-SARS-CoV-2 prophylaxis in healthy subjects. Does it make sense or not? As it turns out, any effects of vitamin C supplementation may be more prominent when the baseline vitamin C level is low, for example in physically active persons. People with hypovitaminosis C are more likely to respond to vitamin C administration. No studies regarding prevention of COVID-19 with high-dose vitamin C supplementation in healthy subjects were found.

## 1. Introduction

Vitamin C is known as L-ascorbic acid since it was noted as the factor needed for the treatment of scurvy (*scorbutus*, lat., i.e., “anti-scorbutus”) [1]. Most mammals can synthesize vitamin C from D-glucose in the liver due to the presence of L-gulono-1,4-lactone oxidase. Because of the lack of this enzyme in humans, they are dependent on vitamin C incorporated into the diet [1,2]. The reduced form of vitamin C (ascorbic acid, AA), is a micronutrient of small size. When it loses its electrons in biosynthetic or antioxidant reactions, it is oxidized to dehydroascorbic acid (DHA), which can easily be converted into L-ascorbic acid [1,2]. The fact that both forms are considered to be equally bioavailable is due to the vitamin C content of foods being usually reported as total vitamin C, i.e., the sum of AA and DHA contents. Both physiological and pathological factors alter the transport and metabolic mechanisms responsible for DHA recycling [3].

Functions of vitamin C are probably related mainly to its electron-donating property. Apart from antioxidant function, it participates in the processes involved in collagen synthesis, synthesis of noradrenaline/adrenaline and peptide hormones, carnitine, gene transcription, translation via different mechanisms, elimination of tyrosine and reduction of iron in the gastrointestinal tract [1]. Severe vitamin C deficiency results in scurvy which is related, among other effects, with impaired immunity and individuals with scurvy are highly susceptible to infections such as pneumonia [4,5]. Following Hemilä [5], in the early 1900s, Casimir Funk (who coined the word “vitamin”), noted that an epidemic of pneumonia in Sudan disappeared when antiscorbutic (i.e., vitamin C-containing) treatment was given to the numerous cases of scurvy that appeared at about the same time. The numerous functions of vitamin C in the body meanits deficiency can be associated with pathomechanisms of infectious diseases, including COVID-19 [6], (Carr, Rowe, 2020). The conducted observational studies and RCTs indeed indicate hypovitaminosis C and visible effects of improving the condition in patients with severe course of SARS-CoV-2 infection, undergoing IV (intravenous) ascorbic acid monotherapy [7,8,9].

The role of vitamin C in immune function has been studied in the last 30 years [10]. Its effects have been confirmed in vitro, targeting the functioning of all populations of the immune system. Vitamin C’s effect on the intracellular content of leukocytes, which is more than 10- to 100-fold higher than in plasma, accumulated against the concentration gradient, suggests not only its anti-inflammatory role but also its intervention in the proliferation and differentiation of these cells [4,8,11]. The intracellular content of vitamin C in leukocytes, which is accumulated against the concentration gradient and is more than 10- to 100- fold higher than in plasma, suggests not only its anti-inflammatory role but also its intervention in the proliferation and differentiation of these cells. Following Chaghouri et al. [12], AA inhibited a proinflammatory nuclear transcription factor-ĸB (NF-ĸB) in human epithelial cell line (ECV 304) and human umbilical vein endothelial cells (HUVECs) and suppressed death of granulocytes. However, AA also presented proinflammatory features, stimulating granulocyte chemotaxis, lymphocyte proliferation, interferon production and increasing antibody blood concentration. Moreover, it enhanced fibroblast proliferation and migration, phagocytosis and generation of reactive oxygen species (ROS) by granulocytes. According to the authors, a significant number of studies reported, on the contrary, the pro-oxidant effects of AA [12]. On the contrary, Fumeron et al. [13] have run a prospective, randomized, open-label trial to assess the effects of vitamin C administered orally (250 mg three times per week) for 2 months on well-defined oxidative and inflammatory markers in 33 chronic haemodialysis patients and reported that supplementation had no effect on oxidative/antioxidative stress and inflammatory markers.

Vitamin C has a number of activities that could conceivably contribute to its immunomodulatory effects, such as that due to its ability to donate electrons, it can protects important biomolecules (proteins, lipids, carbohydrates, and nucleic acids) against the effects of oxidants generated not only during normal cell metabolism but also through exposure to toxins and pollutants (e.g., cigarette smoke) [14]. AA tightens the endothelial permeability barrier in basal cells and also prevents barrier leak generated by inflammatory agents which needs nitric oxide derived from activation of nitric oxide synthase [14]. Vitamin C is also a cofactor for a family of biosynthetic and gene regulatory monooxygenase and dioxygenase enzymes [4]. As a cofactor for the two hydroxylases required for stabilization of the tertiary structure collagen, vitamin C affects the skin structure being a barrier against external insults, including pathogens [4]. Due to gene regulation, vitamin C can participate in T-cell maturation [15]. It is involved in regulating the proliferation and functionality of NK cells and B lymphocytes, the main component of adaptive immunity, affecting the serum concentrations of IgA and IgM, protects the integrity of the neutrophil wall, improves chemotactism, migration, microbial phagocytosis and ROS release. At the level of monocytes/macrophages, it inhibits the secretion of cytokines, especially IL-6 and TNF-α, stimulates phagocytosis and clearance of macrophages [11,16]. Sorice et al. [16] cited the studies on the regulation of influenza virus-infected Gulo -/- mice, deprived of L-gulono-ɣ-lactone oxidase (Gulo), the final enzyme needed for the biosynthesis of vitamin C, as they cannot synthesize AA similar to humans. As was noted, AA played a key role in antiviral immune responses against influenza virus through the increase of IFN-IL-1α/β production. However, when Gulo -/- mice were supplemented with AA after virus inoculation, a viral replication suppression was not observed. It was concluded that the sufficient plasma AA levels could effectively prevent in vivo pathogenesis of influenza virus but probably at the initial stage of viral infection. Furuya et al. [17] noted that AA rather weakly inhibited the multiplication of viruses of three different families: herpes simplex virus type 1 (HSV-1), influenza virus type A and poliovirus type 1. Moreover, as it turned out, DHA, without reducing ability showed much stronger activity against viral infections than AA indicating the factors other than an antioxidant mechanism.

Among the immunomodulatory mechanisms in which AA is involved, there are also those that in the current times of the COVID-19 pandemic deserve special attention because they are common to the pathways in which the coronavirus intervenes or strikes. As was shown by Holford et al. [7], vitamin C abolished upregulated ACE2 (Angiotensin-Converting Enzyme-2), present on the surface of many types of cells that was identified as the receptor for SARS-CoV-2 entry via binding of viral Spike (S) glycoprotein in human arterial endothelial cells. According to Kumar et al. [18], magnesium ascorbate has been revealed as, among 106 other nutraceticals, the most powerful inhibitor of M^PRO^/3CL^PRO^, a key protease in SARS-CoV-2, that as a nonstructural protein participates in the replication and life cycle of the virus inside the host cell. Due to its interaction with ACE2, vitamin C is also mentioned as one of the natural compounds having the inhibitory activity against different variants of coronavirus, such as Alpha-, Beta-, Gamma-, Delta-, Kappa- or Mu-coronavirus [19]. As was shown by Ivanov et al. [20], AA and its specific combinations with other natural compounds lower ACE2 expression at the protein level and at the RNA level as was proved in Human Small Airways Epithelial Cells (SAEC) and Human Microvascular Endothelial Cells (HMEC). Goc et al. [19] showed that the defined mixture of plant extracts and micronutrients with vitamin C demonstrated a highest efficacy by inhibiting the receptor binding domain (RBD), required to bind SARS-CoV-2 to ACE2 receptor. The mixture also significantly inhibited other mechanisms of viral infectivity, including the RNA-dependent RNA polymerase (RdRP), necessary for viral replication, as well as furin and cathepsin L, important for fusion of virions with host membranes and involved in viral multiplication. The mentioned mixture was most effective in inhibiting the original SARS-CoV-2 and Delta variants compared to other forms, Alpha-, Beta-, Gamma- and Mu- (Figure 1).

In this work, the known immunomodulatory properties of vitamin C will be presented, due to which, additionally, it could be considered as a dietary supplement effective in preventing against SARS-CoV-2 infection. Research findings that challenge this hypothesis have also been taken into account.

The article was prepared on the basis of a literature review, narrowed to studies published in English. Nearly half of the cited works have been published in the last 3 years. The survey included all available types of the studies on vitamin C activity in the immune system and against SARS-CoV2 infections and the similar viral infection, such as common cold, excluding those that have included the use of supplements other than vitamin C in parallel and in most of those that concerned the therapeutic application of vitamin C during COVID-19 infection, sepsis or pneumonia.

## 2. Vitamin C and Its Mechanisms of Immunomodulation

The most of the immodulatory effects exerted by vitamin C are due to: the impact on leukocyte functions, gene regulation and being the enzyme cofactor.

### 2.1. Vitamin C and Leukocyte Function

Neutrophil activation and infiltration into infected tissues is an early step in innate immunity [4,21]. These are one of the first cellular responders to the inflammatory signals. Their dominant function is to destroy invading microorganisms and thereby inhibit systemic infection. They phagocytose microorganisms and subject them to an array of killing mechanisms, nonoxidative or oxidative [22].

The primary role of vitamin C in modulating the immune response is due to its indirect influence on the mobility and function of leukocytes, mainly neutrophils. The regulatory mechanisms of this vitamin are the result of its function as an enzyme cofactor and regulator of gene expression [2,4,23]. The role of vitamin C in leukocyte functions is evidenced by the fact that neutrophils, but also phagocytes and T- and B-lymphocytes, accumulate significant amounts of vitamin C due to the active intracellular transport with the participation of sodium-dependent vitamin C transporters (SVCTs) against a concentration gradient, which results in 50 to 100 times higher values than plasma concentrations [4,23]. Neutrophils engage SVCT co-transporter, SVCT2 to accumulate vitamin C and contain intracellular levels of at least 1 mM [2,4]. During oxidative stress, neutrophils can absorb vitamin C also in the oxidized form, DHA through the nonspecific uptake, via glucose transporters (GLUT) [24]. DHA reduction to AA gives about 10 mM of vitamin C inside the cells [2]. DHA being reduced to AA allows further influx of DHA that results in increases of intracellular ascorbate by up to 20-fold [24]. Antioxidant properties of vitamin C define its role for leukocyte function [25]. The excessive production of ROS, following, for example, exercise may damage neutrophils and impair their function, potentially contributing to immunosuppression [25]. Accumulation of milimolar concentrations of vitamin C into neutrophils, particularly following activation of their oxidation burst, is thought to have a protective role in the case of oxidative damage [4]. According to Webb and Villamor [25], supplementation with vitamin C may protect against exercise-induced damage by ROS and preserve the functional ability of neutrophils. However, as was presented by the authors, during small, randomized, cross-over studies, supplementation with 1000 mg/day–1500 mg/day of vitamin C for 7, 9 or 14 days prior to the exercise challenge or to apnea diving (repeated episodes of hypoxia and reoxygenation inducing oxidative stress), dependently on the study, resulted in the significant higher neutrophil counts after exercises or significantly decreased exercise-induced increases of leukocytes and neutrophils compared to placebo or had no effects on neutrophil, monocyte, NK cells or total leukocyte counts after 12 h or the 2.5 h running challenge. The variation between the results are explained, among other factors, as being due to: too relatively small sample size; the types of exercise challenge; the duration of supplementation; and the intake of other nutrients during the trials [25]. Anderson and Theron [26] observed the improved neutrophil motility in response to the endotoxin-activated serum (EAS) and partially purified human recombinant polypeptide having an imperative role in chemotaxis (C5a) in patients with recurrent bacterial infections when patient’s neutrophils were incubated with 5 × 10^−2^ M calcium ascorbate or 10^−1^ M sodium ascorbate in the presence of 5% autologous serum in vitro. Six of the patients were supplemented with oral ascorbate, i.e., 1 g/day for four children and 3 g/day for two adults, and tests of neutrophil migration were performed at monthly intervals thereafter. Improved neutrophil motility was observed in all patients and this correlated with clinical improvements in five of them.

Neutrophils express more than 30 different chemokine and chemoattractant receptors to respond to tissue damage signals [4]. Activated cells release ROS, proteolytic enzymes and proinflammatory mediators that can damage tissues and prolong the inflammatory process. Apoptosis and subsequent phagocytosis of cell debris are the desired effects for the removal of unnecessary neutrophils to clear sites of inflammation. Quenched neutrophil apoptosis has been reported in patients with sepsis and it seems to be related to disease severity [4,7,23,24]. Neutrophils that fail to undergo apoptosis instead undergo necrosis and the subsequent release of intracellular components, such as proteases, can deepen tissue damage [23]. When caspases are inactivated, then necroptosis (NETosis)—the form of neutrophil death is activated [23]. Necroptotic signaling pathways can induce the release of neutrophil extracellular traps (NETs), formed of neutrophil DNA, histones and enzymes. NET-associated histones can stimulate the immune system and generate further tissue damage [23,27]. The production of NETs is a form of cell death which is different from apoptosis and necrosis [22]. As was shown by Mohammed et al. [27], vitamin C sufficiency attenuated NETosis in septic mice and vitamin C deficient polymorphonuclear neutrophils were more susceptible to undergo NETosis through the increased NF-ĸB activation and indirectly through modification (hypercitrullination) of histones and chromatin decondensation. According to Ferrada et al. [2], receptor-interacting serine/threonine-protein kinase 1 (RIPK1) is the key protein that regulates mechanisms of apoptosis and necroptosis. It is known that RIPK1 can be inhibited, besides ubiquitination also via phosphorylation by IĸB kinase complex (IKK α/β), and this process is independent of NF-ĸB activation. In this way, inhibited RIPK1 favors cell survival. The authors suggest that the forms of ascorbic acid, oxidized, DHA and reduced, play a role in the signaling pathway regulating the choice of the type of cell death, apoptosis or necroptosis. AA has been reported as an inhibitor of apoptosis through decreasing of the caspase activity and the expression of BAX, increasing the levels of Bcl-2 and preventing the release of cytochrome C from mitochondria [2]. On the other side, vitamin C may be expected to protect the oxidant-sensitive caspase enzyme and the same apoptotic pathways [2,4,23]. Following Ferrada et al. [2], vitamin C can act as an electron donor for the reduction of cytochrome C. The reduction mediated by AA results in its oxidation which could generate oxidative stress and in generation of DHA. Thus, vitamin C could play a dual, i.e., protective or destructive, role in mitochondria. The protective role would work under physiological doses of AA where it would be constantly reduced without influencing the mitochondrial redox potential. In high doses, the destructive role could be activated where the reduction of cytochrome C would favor the oxidation of AA to DHA which would result in cell death due to mitochondrial stress [2]. The accumulation of DHA results also in the inhibition of IKK α/β and p38 which can trigger the activation of RIPK1 and necroptosis. The authors suggest a possible scenario that DHA primarily targets the activation of RIPK1 to prevent its inhibitory phosphorylation and AA as an inhibitor of apoptosis probably preconditions cells to necroptosis [2]. As was shown by Mohammed et al. [27], Vitamin C-deficient Gulo (-/-) mice were i.p. injected with a fecal stem solution to induce abdominal perionitis (FIP) 30 min prior to receiving AA (200 mg/kg). At a cellular level, the Vitamin C deficient polymorphonuclear neutrophils (PMN) displayed: the increased expression of PAD4 mRNA, i.e., peptidylargininedeiminase, participating in decondensation of nuclear chromatin by the removal of arginine residues on histones; induction of autophagy, as one of the type of cell deaths required for necroptosis; attenuated apoptosis and the endoplasmic reticulum stress (ER stress) which is related with autophagy and ultimately contributes to the cell fate decision. All these effects were quenched in the Vitamin C-sufficient Gulo (-/-) mice.

With the development of imaging technology, it has been found that the recruited polymorphonuclear leukocytes (PMNs) could migrate back to the circulation and it is a new way of PMN clearance in inflammatory or injury site. This process without undergoing apoptosis has been called as “PMN reverse migration (rM)” [28]. In 2004, Sharma et al. [29] showed that after treatment of PMNs with *E. coli* or arachidonic acid, AA or DHA did not influence the phagocytosis of PMNs but improved ROS generation and apoptosis. ROS generation was potentiated due to nitric oxide (NO), the synthesis of which was enhanced by ascorbate. Parker et al. [14] suggested that the increased synthesis of NO and the neutralization of ROS or NOS by ascorbate is associated with the tightening of the endothelium permeability barrier by stabilizing microtubules.

Research on the influence of vitamin C on the functionality of neutrophils in the inflammatory processes, including those caused by infection, is not conclusive. On the basis of a prospective, randomized, double-blinded clinical trial with placebo group (*n* = 10) and vitamin C-treated group (*n* = 10), Ferrόn-Celma et al. [30] noted that 450 mg/day for a 6-day period given to the septic abdominal surgery patients was reflected in a nonsignificant reduction in Fas (CD95) expression on CD-15-positive peripheral blood neutrophils, significantly decreased caspase-3 and PARP levels and significantly increased Bcl-2 levels. Peripheral blood samples were obtained daily from 24 h after vitamin C administration until 6 h postsurgery. All the above events had an antiapoptotic effect on peripheral blood neutrophils, although not at all time points.

### 2.2. Immunomodulation and Vitamin C as an Enzyme Cofactor

The enzymatic roles of vitamin C are related to dioxygenases or monooxygenases. All these enzymes have a metal, iron or copper in their active sites and the mechanism of vitamin C action seems to be related to reduction or maintenance of these metals in the reduced state [1]. However, in some cases there is direct involvement and in others, vitamin C can recover the enzymatic function if the metal atom is oxidized [1]. As was suggested by Doseděl et al. [1], it could potentially be that other reductants in place of vitamin C are less effective.

The largest group of enzymes having vitamin C as one of their cofactors is the iron-dependent and 2-oxoglutarate-dependent dioxygenases superfamily (2OGO) and, as was shown, this vitamin is also the specific cofactor of hydroxylases involved in the regulation of the cellular sensor hypoxia-inducible factor 1α (HIF1 α), e.g., prolyl-hydroxylase domain-containing proteins (PHDs) and asparaginyl hydroxylase termed factor inhibiting HIF (FIH) [1,31,32,33]. Oxygen-dependent PHD activity leads to degradation of HIF-1α via a ubiquitin ligase complex regulated by the von Hippel–Lindau (VHL) protein (pVHL), elongin B and C, Cullin 2 and RING-box 1 proteins [33,34,35]. Three isoforms of PHD have been characterized, termed PHD1, PHD2 and PHD3, different in size, subcellular localization and tissue distribution [32]. According to Kuiper et al. [34], the K_m_ for ascorbate has been shown as 140–180 and 260 µM for the PHDs and FIH, respectively. This suggests that FIH could be more susceptible to ascorbate loss than PHDs and implies that ascorbate may be more effective at preventive HIF-1 transcriptional activity, which is regulated by FIH than protein accumulation, regulated by the PHDs [34]. As reported by Bozonet et al. [22], NETs’ formation may be dependent on HIF-1. HIF-1α regulates the transcription of many genes related to the control of metabolism, cellular stress response and cell survival [22]. It is downregulated by iron-containing 2-oxoglutarate-dependent enzymes (abovementioned hydroxylases) having ascorbate as a cofactor [22]. As was mentioned by Ji and Fan [28], some studies revealed negative regulation of PMN rM with the suppressing role of HIF-1α, noted in zebrafish. Elks et al. [33] with use of indirect HIF inhibitor, dimethyloxaloyglycine (DMOG) revealed that activated HIF-1α delays resolution of inflammation and it is a consequence of reduced neutrophil apoptosis and increased retention of these cells at the site of injury. Kuiper et al. [34] showed that ascorbate inhibited HIF-1 activity and HIF-1-dependent gene expression. In the experiments with hydroxylase inhibitors, such as CoCl_2_, NiCl_2,_ desferrioxamine, dimethyloxalylglycine and hypoxia, ascorbate inhibited HIF-1 activity most significantly with all mechanisms of iron competition. HIF-1-dependent gene expression was effectively prevented by ascorbate and was inhibited even under conditions that allowed HIF-1α protein stabilization. On the basis of their results, the authors suggested that ascorbate acts dominantly to stabilize and reduce the iron atom in the hydroxylase active site and that the asparagine hydroxylase which regulates HIF-1 transcriptional activity is particularly susceptible to fluctuations in intracellular ascorbate. According to the same authors [34], ascorbate is able to prevent HIF-1 transcriptional activation at the gene promoter. In human endometrial adenocarcinoma cell line (Ishikawa cells), prosurvival protein BNIP3 production was decreased by ascorbate preloading, remaining at the control levels with all the mechanisms of HIF induction, despite significant levels of HIF-1α present in the same cells. Moreover, ascorbate-loaded human T-lymphocytes (Jurkat cells) and human umbilical endothelial cells (HUVEC) had significantly decreased BNIP3 levels in response to all HIF-1 inducers used, with the exception of HUVECs at 1% O_2_, in which ascorbate did not affect BNIP3 expression.

### 2.3. Immunomodulation and Vitamin C as a Regulator of Gene Expression

The role of vitamin C as an enzyme cofactor involves, apart from regulating enzymatic activity, influencing gene expression. Although not all reports show this unequivocally. Vitamin C as a cofactor for iron- and 2-oxoglutarate-dependent enzymes regulates hydroxylation of the methylated moieties in DNA; for example, it is a cofactor for the ten–eleven translocation (TET) dioxygenases hydroxylating methylated cytosine groups in DNA and for Jumonji C domain-containing histone demethylases (JHDMs) that catalyze histone demethylation by prior hydroxylation of mono-, di- and trimethylated histone lysine and arginine residues [23,36]. Both groups of enzymes, i.e., TET and JHDs, are included in epigenetic mechanisms involved in regulation of T-cell maturation [4,23]. As was shown by Manning et al. [15] in the studies on the lymphocyte progenitor cells, supplementation of a basal culture medium MEM with AA or with its stable phosphate derivative was necessary for maturation of T-cell receptor αβ (TCRαβ) with surface antigens CD4 and CD8 and that process was partially due to regulation of *CD8A* gene expression by methylation or demethylation of histone proteins and of CpG motifs in regulatory regions of DNA. Enzymatic removal of methyl groups was mediated in part by a superfamily of iron- and 2OGO, including members of the Jumonji C histone-lysine demethylase family. Under the influence of AA, the authors also noted upregulation of *ZAP70,* the gene encoding the kinase that is key to signal transduction via TCRαβ complex. Following Mohammed et al. [21], due to the influence of vitamin C on gene expression, macrophage functions are modulated and the vitamin’s participation in suppressing inflammation and sepsis is related to this mechanism. During inflammation, the mobilized macrophages are divided into three categories: activated, which secrete proinflammatory factors; alternatively activated, considered as anti-inflammatory; and regulatory which secrete considerable amounts of anti-inflammatory cytokines. To address whether the modulatory activities of vitamin C are effective in human monocyte/macrophages, Mohammed et al. [21] exposed human acute monocytic leukemia cells (THP-1 cells) to bacterial lipopolysaccharide (LPS) to examine the mRNA expression of the proinflammatory genes: *IL-6*, *IL-8* and *TNF**α*. The authors increased intracellular concentrations of vitamin C by treatment of cells with AA prior exposure to LPS. LPS exposure of untreated THP-1 cells resulted in a robust activation of mRNA of these genes. Treatment of cells with AA did not affect the baseline proinflammatory gene expression. However, LPS exposure of AA-treated cells resulted in significant attenuation of mRNA for all three cytokines. This observation is consistent with the results obtained by Canali et al. [10] during a pilot study on peripheral blood mononuclear cells (PBMNC), isolated from healthy volunteers, that supplementation of vitamin C in healthy subjects (1 g/day for 5 days) having a sufficient vitamin C plasma concentration at the baseline, did not result in a significant modification of gene expression profile. However, following an inflammatory factor such as LPS, the higher availability of AA emerged and resulted in a significant modulation of cell response. As was shown by the authors [10], after 5 h incubation of PBMNC with LPS, some genes were induced before AA supplementation that was associated with the activation of NFĸB and MAPK signaling cascades which was reflected in the increase of the expression of mRNA encoding for proinflammatory cytokines, such as TNFα, IL-1B, IL-1A, IL-6 and IL-8. After AA supplementation, the insignificant decrease in TNFα mRNA transcription was noted. In turn, the presence of vitamin C in PBMNC was associated with an early activation of anti-inflammatory cytokine IL-10 synthesis. Following the authors, IL-10 can limit or terminate the inflammatory responses via inhibiting proinflammatory cytokine production either as mRNA accumulation or protein release [10]. In the study performed by Härtel et al. [37], vitamin C dose dependently inhibited the LPS-induced number of human monocytes producing IL-6 (e.g., 41% reduction, *p* < 0.001, 20 mM vitamin C) and TNFα (e.g., 26% reduction, *p* < 0.005, 20 mM vitamin C). Simultaneously, the number of human lymphocytes producing IL-2 after phorbol 12-myristate 13-acetate (PMA)/ionomycin (the mixture optimally stimulating cytokine production) [38] treatment was dose-dependently reduced (e.g., 24.2% inhibition, *p* < 0.005, 20 mM vitamin C). The number of IL-1 and IL-8 producing monocytes as well as TNFα and IFN-ɣ producing lymphocytes were not significantly affected by 20 mM vitamin C [37]. The authors suggested three possible mechanisms evoked by vitamin C which made the changes in the cytokine profile. The first, related to NF-ĸB, inappropriately activated by oxidative stress triggered by LPS-induced sepsis or by a number of inflammatory stimuli, including TNFα and IL-6. The second, via activation of the p38 mitogen-activated protein kinase and the third, by inhibiting T cell apoptosis signaling pathways and FAS-induced apoptotic death in human monocytes. The authors emphasized the selective influence of vitamin C on cytokines [37]. On the contrary, Żychowska et al. [39] investigated the effects of supplementation and physical exercise on the expression of genes related to immune response, such as *CCL2*, *CRP*, *IL-1*; *IL-6* and *IL-10* in women. The clear tendency of a decrease in IL-6 and an increase in IL-10 mRNA was observed in leukocytes from the training women supplemented for 6 weeks with 1000 mg of vitamin C as referenced to the control group receiving placebo. Changes in IL-1 and CCL2 mRNA expression were noted only in the control group and were probably associated with training. CCL2 ((C-C motif) ligand 2) is a chemokine capable of recruiting monocytes, memory T-cells and dendritic cells to induce pro-inflammatory response. As stated by the authors, the hypothesis that supplementation with vitamin C in a dose of 1000 mg/day can cause a decrease in proinflammatory and an increase in anti-inflammatory gene expression has not been proved in this trial. No significant changes in gene expression of all studied genes were noted in the supplemented group.

## 3. Supportive Supplementation—Does High-Dose Vitamin C as a Dietary Supplement Used in Prophylaxis for Anti-SARS-CoV-2 in Healthy Subjects Make Sense?

Interest in vitamin C in the context of the prevention of respiratory tract infections was begun by Linus Pauling, who reported decades ago that a daily intake of 1 g can reduce the incidence of colds by about 45% and the optimal daily intake of vitamin C to prevent disease should be at least 2–3 g [40,41]. Other studies conducted later were not and still are not unambiguous.

In view of the constant search for therapeutic options or options supporting COVID-19 therapy, especially ones that are easily available and cheap, the area of interest and each obtained research result are valuable. Potentially, dietary supplementation could be one of the options to support the body’s protection against viral infection. Among the substances whose mechanisms of action justify the immune potential and specific effectiveness in protection specifically against coronaviruses, one can indicate lactoferrin (LF) and vitamin D [42,43,44]. LF is a multifunctional glycoprotein, isolated from human milk, a component of the innate immune response and a potent immunomodulator and anti-inflammatory agent [43]. Some authors attribute to it a preventive role through interaction with viral receptors or the upregulation of the antiviral response of the immune system rather than inhibiting virus replication in the target cell [45,46]. Sinopoli et al. [43] during the systematic search provided data on the effects of orally administered LF in the prevention and/or management of viral infection caused by *Flaviviridae*, *Retroviridae*, *Coronaviridae*, *Reoviridae* and *Caliciviridae*. Despite that the findings were heterogenous across and within viral families and the clear conclusions were not possible to draw, some positive results were reported about decrease of symptom severity and duration or reduction in viral loads. Similarly, the need for well-planned follow-up research is shown in the case of vitamin D [44]. It is believed to have various immune-regulatory roles including: promoting anti-inflammatory or downregulation of proinflammatory cytokines; blocking entry and replication of SARS-CoV-2 or the production of antimicrobial peptides [47]. There are also indications that lead us to believe that low vitamin D status is a risk factor for COVID-19 disease and poor outcomes, although it is not possible to confirm causality because many risk factors are common to hypovitaminosis D and the severe course of COVID-19. However, although the recommended daily dose that seems to maintain the sufficient serum level of this vitamin, i.e., 400 UI, is known, whether this is enough to gain any benefits for COVID-19 patients or prevention remains unclear [42,44,47]. The leading health bodies (Public Health England, the National Institute for Health and Care Excellence and the Scientific Advisory Committee on Nutrition) highlight the need for further research to provide evidence on vitamin D and its beneficial effects on COVID-19, especially high-quality RCTs [42,44].

Vitamin C is the next vitamin that can be considered as a supplement supporting the body in the fight against viral infection. The known functions that vitamin C performs in the immune system warrant further research into the need for vitamin C supplementation in the body. The only question is obvious, i.e., whether we should provide ourselves with only the right level of it and will it be enough, or do we need much more in times of a pandemic?

Following Cerullo et al. [40], independently, on the confirmed role of vitamin C in the immune system, the preventive role of dietary supplements based on vitamin C and, the same, reduction of the incidence of viral disease in a well-nourished population are not supported in literature. Despite this, the belief that an extra intake of vitamin C can increase the efficacy of the immune system is still popular. Otherwise, it is fairly common in the present day of the pandemic that high oral doses of vitamin C, i.e., significantly exceeding Recommended Daily Allowance (RDA) values, are effective in protecting against SARS-CoV-2 infection in healthy subjects.

When reading the literature related to this topic, one can sometimes get the impression that the authors, when writing about prevention, do not mean preventing understood as a decline in morbidity in general, but preventing the further development of the disease and applying this expression to the situation of alleviating the symptoms and effects of an infection already existing in the body—this should be distinguished.

This chapter discusses the justification of the now relatively popular use of vitamin C as a dietary supplement used in a high dose, significantly exceeding RDA, in the prevention of SARS-CoV 2 infections in healthy subjects. Does it make sense or not?

COVID-19 is a pandemic caused by coronavirus SARS-CoV-2 with mild to severe respiratory symptoms [48]. Coronaviruses and influenza are among the pandemic viruses that can cause lethal lung injuries and death from acute respiratory distress syndrome (ARDS) [11,49]. As was given by Earar et al. [11], three clinical stages of SARS-CoV-2 infection have been described: asymptomatic stage I and II with mild and moderate symptoms, accompanied by the presence of the virus, and stage III with severe respiratory symptoms, including pneumonia, sepsis or multiple organ failure. Preliminary data suggest that COVID-19 pneumonia is a complication caused by the hyperactivation of immune effector cells that, in turn, triggers a cascade of further events, such as the increased oxidative stress which leads to a systemic inflammatory response due to overproduction of cytokines, contributing to ARDS. Oxidative imbalance results in oxidation of reactive residues of redox-sensitive proteins [50,51]. As was proven by Hoang et al. [51], coronaviruses and influenza induced significant downregulation of the antioxidant system that results in lethal lung injury and death from ARDS due to oxidative damage. According to Feyaerts, Luyten [50], many patients who are severely ill with COVID-19 have elevated cytokine levels, including IL-6, resembling the cytokine storm described in SARS and the Middle East respiratory syndrome (SARS-CoV and MERS-CoV, respectively). Because of this, it is possible that high mortality is due to virus-driven hyperinflammation. In the light of this, considering the role of vitamin C in COVID-19 therapy and/or prophylaxis is not pointless. The influence of vitamin C on the immune system and participation in anti-inflammatory processes predestined this compound to be a potentially anti-infective vitamin. Moreover, the intensified inflammatory processes consume the available antioxidants, including vitamin C, and the body’s demand for substances with such properties increases significantly [51]. As reported by Atherton et al. [52], vitamin C specifically targeted coronaviruses. The authors noted that chick-embryo ciliated tracheal organ (CETO) cultures previously exposed to AA exhibited considerably increased resistance to infection by coronavirus (infectious bronchitis virus, IBV) while they did not affect the resistance of cells to infection by the orthomyxovirus (influenza) or the paramyxovirus (Newcastle disease virus, NDV).

Following Hemilä and de Man [53], a recent survey found that out of 18 COVID-19 patients, 17 had undetectable vitamin C levels and 1 patient had a very low level of that vitamin. Another study also reported low vitamin C plasma levels in COVID-19 patients and nonsurvivors had half the plasma level of survivors [53]. Scurvy has long been associated with pneumonia that suggests the influence of vitamin C on susceptibility to respiratory infections [54]. As was reported by Rosetti et al. [9] who analyzed clinical trials and research papers, 40% of critically ill patients with sepsis have serum vitamin C levels that suggest scurvy (<11.3 µM/L). Arvinte et al. [55], in a pilot study conducted in a cohort of critically ill COVID-19 patients of an ICU, reported that most patients were deficient in vitamin C and D and asked a question inter alia: “Should those at risk for or newly diagnosed with, SARS-CoV-2 infection have their serum vitamin C and vitamin D levels measured, and started on pre-emptive supplementation to lower risk of COVID-19, and severe forms?”.

All the above examples point to the undoubted relationship between vitamin C and infections. However, the functions of vitamin C cannot be viewed equally during use in therapy and as a dietary supplement in healthy subjects. Therapeutically, vitamin C participates in advanced processes in an already infected organism, and prophilactically it should prepare the immune system to the increased infection threat. When analyzing the reports on the beneficial effects of vitamin C recorded in patients, e.g., with advanced sepsis or in the cases of pneumonia, mostly we deal with a different route of administration than when dietary supplementation administered only *per os* is used [7]. In hospital conditions, vitamin C IV is used as a drug supporting the body in the fight against rapid inflammation caused by, for example, sepsis. Therapeutic use of high doses of IV ascorbic acid, most often administered to the patients with sepsis or ARDS that are also the consequences of COVID-19, is related to the simultaneous low levels of this vitamin determined in plasma [9,56]. IV ascorbate can achieve to 10 or 25 times higher serum levels than when the same amount in ingested [57]. It needs to be remembered that levels in serum following oral administration are limited by, among others, gastrointestinal absorption, renal function and metabolic rate [57]. Padayatty et al. [58] developed the 3-compartment vitamin C pharmacokinetic model that predicted plasma vitamin C concentrations after i.v. or oral administration. During 3 g given orally every 4 h, predicted peak plasma concentration was approximately 220 µM/L. After i.v. administration peak plasma vitamin C concentrations were: 1760 µM/L for 3 g; 2870 µM/L for 5 g; 5580 µM/L for 10 g; and 15,380 µM/L for 100 g. Simultaneously, predicted peak urine vitamin C concentrations were 140-fold higher after parenteral (i.v.) administration compared with oral administration. So, considering the efficacy of vitamin C oral supplementation, the aspect of vitamin C bioavailability should be taken into account. It should be also considered what doses should be defined as having the desired effect for the specific purpose adopted, such as prophylaxis, in relation to doses defined as providing the recommended daily intake for the proper functioning of the organism in general. The recommendations for vitamin C daily intake can be summarized into four main groups, such as: preventing scurvy; partly saturating immune cells with vitamin C with limiting its urinary excretion; maintaining adequate plasma vitamin C status; and to optimize health by ingestion of the optimal amount of vitamin C [59]. It would seem that the values defined by experts as RDA take into account the amount of vitamin that provides balance in the body, including the proper functioning of the immune system. The point is: does the potential beneficial effect positively correlate with the dose which would justify supplementation of healthy people with doses exceeding the RDA?

According to Carr and Lykkesfeldt [59], the current global RDAs for vitamin C vary by ca. 3-fold, dependently on the regions of the world, from 40 mg/day in the UK and India to 110 mg/day in the European countries. According to the European Food Safety Authority (EFSA) [60], Daily Reference Values for vitamin C were defined in the range from 45 to 90 mg/day for adults and between 30–40 mg/day for children, depending on the gender and age and the source of the values determination, i.e., The Institute of Medicine (IOM), USA or WHO/FAO. The basis for deriving the reference values for vitamin C intake was, among others: a bioavailability at an intake of 100 mg vitamin C/day, the metabolic loss (2.9% per day) and compensation for the urinary loss (25% per day) [59,61]. Additional intake as 35 mg/day is recommended for smokers (IOM, 2000). The German, Austrian and Swiss nutrition societies (The D-A-CH) suggest even 100 mg more of vitamin C for smokers, i.e., 135 mg/day for female smokers and 155 mg/day for male smokers [61]. As was given by Carr and Lykkesfeldt [59], in 2006 Australia and New Zealand developed an additional reference value, i.e., the suggested dietary target (SDT) with chronic diseases prevention in mind. The SDT for vitamin C for healthy men and women was defined to be 220 mg/day and 190 mg/day, respectively. Similarly in China, where a proposed intake (PI) was adopted to prevent or reduce the risk of noncommunicable diseases. The PI target of 200 mg/day was defined as saturating measured as the plasma concentration 70 µM/L. All these values proposed or suggested for preventive action exceeded RDAs, defined for nonsmokers, at least twice. According to Kuiper et al. [34], the circulating plasma concentration of 70–80 µM/L ensures the maintenance of vitamin C tissue levels. Following Holford et al. [7], the plasma level ca. 50 µM/L is sufficient to prevent scurvy but may be not sufficient under viral exposure and physiological stress. Indeed, the same authors cited the study from New Zealand that patients with pneumonia had much lower vitamin C levels than healthy controls (23 µM/L vs. 56 µM/L, *p* < 0.001) [7]. The increased consumption of ascorbic acid in disease states, as was mentioned earlier, justifies the need for much higher intakes of vitamin C during, for example, viral infections, i.e., 2–3 g/day to maintain normal plasma levels (60–80 µM/L) [7]; but what about real prophylaxis?

As was shown by Webb and Villamor [25], in two RCTs vitamin C supplementation at 2000 mg/day for 2 months was reflected in an 85% reduction of the incidence of pneumonia in US military recruits compared with placebo. In turn, in a randomized, placebo-controlled and double-blind trial of soldiers in the Korea Army Training Center with participation of 1444 persons, administration of 6000 mg/day of vitamin C for 30 day resulted in an 8.4% risk reduction for catching a cold [62] (the results were corrected by Hemilä and Chalker [63]). Despite these results, further studies described in this paper suggest that the principle: “the more, the better” does not apply to the preventive doses of vitamin C. According to the Swiss expert panel and the Swiss Society of Nutrition, the target for supplementation with 200 mg of vitamin C is to strengthen the immune system [7]. Pharmacokinetic studies in healthy volunteers prove a 200 mg daily dose to produce a plasma level of ca. 70 to 90 µM/L and in the light of the available data, mentioned earlier, it seems to be sufficient for the immunological protection of the balanced organism [34]. Complete bioavailability and saturation of the leukocytes with vitamin C ingested at the dose of 200 mg much earlier has been confirmed by Levine et al. [64], on the basis of the experiment with seven healthy volunteers who were hospitalized for 4–6 months and consumed a diet containing less than 5 mg or from 30 to 2500 mg in seven daily doses of vitamin C. The authors reported as follows: neutrophils, monocytes and lymphocytes saturated at 100 mg daily and contained concentrations 14 times higher than plasma. Bioavailability was complete for 200 mg given as a single dose. No vitamin in urine excretion was noted until the 100 mg dose. At single doses equal to or above 500 mg, bioavailability declined and the absorbed amount was excreted. In healthy subjects, mostly consistently with Levine et al. [64], Cerullo et al. [40], based on the individual variability, suggested that a daily intake of vitamin C between 100 mg and 400 mg ensures 100% of the bioavailability and blood saturation with a steady state of plasma concentration that reaches a maximum level of ca. 70–80 µM/L. According to the authors, if the intake of vitamin C is more than 500 mg/day, a further increase in plasma concentration is inhibited and the bioavailability can decrease even to 30% when one dose of vitamin C exceeds 1000 mg. This occurs because when 500–1000 mg of vitamin C is administered orally, the transporter SVCT1 rapidly achieves its maximal saturation, while urine excretion of the vitamin is progressively increased [40]. Moreover, according to Hemilä and de Man [52], based on the dose vs. plasma level analyses, it is unlikely that a healthy person would benefit from daily vitamin C doses higher than 500 mg/day. At 1000 mg/day, oxalate and uric acid excretion were elevated. On the basis of these results, the authors proposed upper safe doses of vitamin C as below 1000 mg/day. Values of 1000 mg–2000 mg/day have been suggested as critical limits also by some countries, based on a potential risk of osmotic diarrhea and related gastrointestinal disturbances in some people at doses higher than these [59].

Following Padayatty et al. [58], the peak plasma vitamin C concentrations increases with i.v. doses, whereas peak plasma vitamin C concentrations seem to plateau with increasing oral doses. According to the authors, vitamin C plasma concentrations are tightly controlled after ingestion even at the highest tolerated amounts, such as 3 g. The authors showed that gram doses produce transient peak plasma concentrations that at most are 2- to 3-fold higher than those from vitamin C-rich foods (200 or 300 mg daily). In either case, plasma values returned to similar steady-state concentrations in 24 h. As the authors claim, differences in plasma concentrations from supplements and from food intake are not large, and because of that, supplements would be expected to confer little additional benefit. According to the German Nutrition Society, vitamin C is an additive in many processed foods, such as meat and sausage products and these foods are also the source of vitamin C. Furthermore, foods fortified with vitamin C are available. In the light of the above, nutritional supplements to ensure a sufficient vitamin C supply are not necessary [61]. Canali et al. [10] conducted a study in which five volunteers were supplemented with 1 g/day of vitamin C for 5 consecutive days. After that time or before supplementation, the gene profiles related to the inflammatory processes were analyzed in peripheral blood mononuclear cells. As was noted by the authors, vitamin C supplementation resulted in a different modulation of gene expression but only under the inflammatory stimulus, LPS which undermined the preventive importance of this vitamin. The authors suggested that vitamin C supplementation in healthy subjects, not selected according to a specific genetic profile and having a sufficient vitamin C plasma concentration at the baseline, did not cause significant changes in gene expression profile. According to the authors, under this satisfactory micronutrient status, supplementation wit 1 g of vitamin C is not immediately evident and is “buffered” within a homeostatic physiological equilibrium. The work by Canali et al. [10] proves the expected effect of high-dose vitamin C in the course of inflammatory processes that are already taking place and not as a prophylactic factor in a healthy body.

Additionally, Heimer et al. [65] presented a critical evaluation of the evidence concerning the efficacy of high-dose vitamin C for the prophylaxis and treatment of the common cold, which is one of the most widespread viral upper respiratory tract infections (URTIs), caused by, among others, coronaviruses. The authors analyzed published clinical trials, literature reviews, meta-analyses and systematic reviews. As they concluded, vitamin C did not prevent the common cold even at a dose of 1 g/day for several months. The exceptions were marathon runners, skiers and soldiers, training in subarctic conditions who may develop an immune stress condition and who were supplemented with minimum 200 mg/day. Strenuous exercise and physical stress affect the immune system by stimulating Th2 immune response and ROS produced during physical stress can impair the motility and functional ability of neutrophils, the first line of defense against viruses [25]. In turn, the elevated production of ROS increases cellular consumption of vitamin C that implicates the decrease of vitamin C levels in plasma [5]. Then, compared to the placebo group, athletes had a relative risk of 0.50 (95% CI 0.38 to 0.66) of developing the common cold. Similarly, Hemilä and Chalker [63] during meta-analysis found that in five trials with 598 physically active participants vitamin C decreased common cold risk by 52% (*p* < 0.00001). According to Hemilä [5], another group in which the beneficial effects of vitamin C as an prophylactic agent may be more prominent consequently brings together people with vitamin C deficiency. The same authors undertook to justify the recommended prophylactic doses of vitamin C and they noted that, when vitamin C intake is below 0.1 g/day, there is a steep relationship between plasma vitamin C level and the dose of the vitamin, and at 0.2 g/day, the vitamin C level in the plasma of people in good health becomes saturated at about 70 µM/L. On this basis, the authors suggested that when healthy people have a dietary intake of about 0.2 g/day of vitamin C, there is usually no reason to expect a response to additional vitamin C supplementation [5]. This conclusion is consistent with the earlier mentioned authors who indicated 200 mg as a dose that guarantees complete bioavailability of vitamin C [64]. Hemilä and de Man [53] or Heimer et al. [65] rightly pay attention to the fact that ascorbic acid as a highly water-soluble vitamin is characterized by rapid clearance of excess levels by the kidneys and, in such cases, it would not be absorbed anyway. Consistently with this, Hoang et al. [51] suggested further investigation on prophylactic ability of vitamin C supplemented rather in low doses.

## 4. Summary

The increased consumption of ascorbic acid in disease states related to the development of the inflammatory processes, as was mentioned earlier, justifies the application of high-dose vitamin C during sepsis or pneumonia resulting also from severe infections. Most of the cases described in the literature where the beneficial effects of vitamin C are shown for various reasons are associated with its deficiency. This agrees with Carr and Rowe [6], who claimed that people who already have hypovitaminosis C are more likely to respond to vitamin C administration. Milani et al. [54] summarized the current evidence regarding the use of vitamin C in the prevention or during therapy of patients with SARS-CoV2 infection, based on available publications between January 2020 and February 2021. Overall, 21 publications were included in the review, describing the case-reports and case-series, observational studies and some clinical trials. No studies regarding prevention of COVID-19 with vitamin C supplementation in healthy subjects were found. Some clinical observations reported improved medical condition of patients with COVID-19 treated with vitamin C but such application should not be understood as the prophylactic role of ascorbic acid sensu strico.

As a conclusion, on the basis of the survey of the available literature, the finding by Cerullo et al. [40], that the regular use of high doses of oral vitamin C in healthy subjects does not contribute to reducing the incidence of COVID-19, is still valid.

## Figures and Tables

**Figure 1 nutrients-14-00979-f001:**
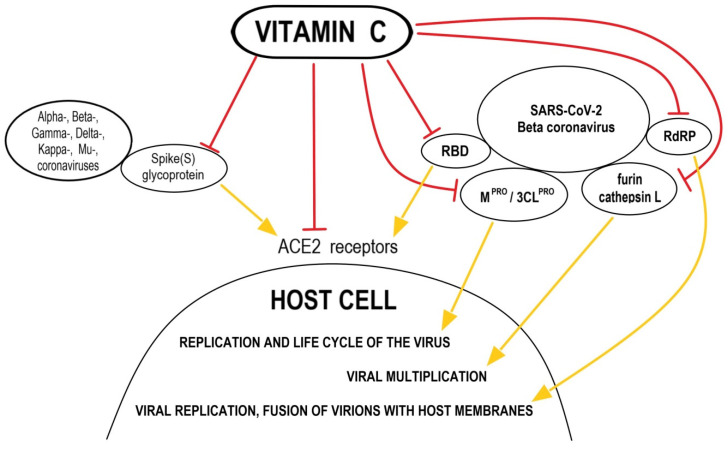
Possible mechanisms of anti-SARS-CoV-2 actions of vitamin C. *Abbreviations*: ACE2—Angiotensin-Converting Enzyme-2; RBD—receptor binding domain; RdRP—the RNA-dependent RNA polymerase; M^PRO^/3CL^PRO^—the key protease in SARS-CoV-2.

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
