# Peer review of "High-Dose Vitamin C Supplementation as a Legitimate Anti-SARS-CoV-2 Prophylaxis in Healthy Subjects—Yes or No?"

_nutrients, 2022, doi:10.3390/nu14050979_

Round 1

Reviewer 1 Report

The manuscript is very informative. However, I have a few concerns which need to be addressed.

  1) Authors need to include the pictorial mechanism of action of vitamin C on  SARS-CoV 2.  2) It would be very interesting to add if High – dose vitamin C affect differently to the different  variants such as " Alpha-coronavirus (HCoV‐229E and HCoV‐NL63), Beta-coronavirus (β-CoV)(HCoV- 61 OC43, HCoV-HKU1, SARS-CoV, MERS-CoV and SARS-CoV-2), Gamma-coronavirus and 62 Delta-coronavirus"

Author Response

Point 1. Authors need to include the pictorial mechanism of action of vitamin C on  SARS-CoV 2 ".

Response 1.The Figure 1 presenting the mechanisms of action of vitamin C on SARS-CoV-2 has been added on page 5.

Point 2. It would be very interesting to add if High – dose vitamin C affect differently to the different  variants such as " Alpha-coronavirus (HCoV‐229E and HCoV‐NL63), Beta-coronavirus (β-CoV)(HCoV- 61 OC43, HCoV-HKU1, SARS-CoV, MERS-CoV and SARS-CoV-2), Gamma-coronavirus and 62 Delta-coronavirus".

Response 2. The fragment has been added on page 4.

Reviewer 2 Report

Thank you for the opportunity to review this manuscript.

Vitamin C is well known for its anti-inflammatory and free radical scavenging properties but considerable controversy still exists regarding vitamin C supplementation among various systematic review and meta-analyses, owing to the diversified methodology of included studies. In general, the study gives an interesting multifactorial impression of the different aspects of the vitamin C, but I have the following comments and concerns.

The introduction should better center the topic. I think that it is unfocused. There are already various manuscripts that address the subject. What does this work add to the already existing literature? (“Carr, A. C., & Rowe, S. (2020); The emerging role of vitamin C in the prevention and treatment of COVID-19. Nutrients, 12(11), 3286, Milani, G. P., Macchi, M., & Guz-Mark, A. (2021). Vitamin C in the Treatment of COVID-19. Nutrients, 13(4), 1172).

I recommend to revisit the introduction emphasizing the relationship with COVID-19 and the possible role of vitamin C.

It is necessary to mention some supplements which could have a role in the nutritional supplementation for COVID-19. For example it is demonstrate that lactoferrin is a role in the prevention and management of viral infection. A good reference is “Sinopoli, A, Isonne, C, Santoro, MM, Baccolini, V. The effects of orally administered lactoferrin in the prevention and management of viral infections: a systematic review. Rev Med Virol. 2022; 32( 1):e2261. https://doi.org/10.1002/rmv.2261”.

Also, Vitamin D should be mentioned in the possible role of SARS Cov 2. Good references are “Vimaleswaran, K. S., Forouhi, N. G., & Khunti, K. (2021). Vitamin D and covid-19. Bmj, 372”, “Wise, J. (2020). Covid-19: Evidence is lacking to support vitamin D’s role in treatment and prevention.

Author Response

Point 1. The introduction should better center the topic. I think that it is unfocused. There are already various manuscripts that address the subject. What does this work add to the already existing literature? (“Carr, A. C., & Rowe, S. (2020); The emerging role of vitamin C in the prevention and treatment of COVID-19. Nutrients, 12(11), 3286, Milani, G. P., Macchi, M., & Guz-Mark, A. (2021). Vitamin C in the Treatment of COVID-19. Nutrients, 13(4), 1172).

I recommend to revisit the introduction emphasizing the relationship with COVID-19 and the possible role of vitamin C.

Response 1. I fully agree with the Reviewer and, at the same time, thank you very much for the remark that in the Introduction, the importance of vitamin C in COVID-19 was lost.

The idea of the article was to discuss the sense of taking high doses of this vitamin by healthy people (which is practiced) in the light of the known mechanisms of vitamin C, before SARS-CoV-2 infection occurs, i.e. only for prophylactic purposes. A general shedding of the role of vitamin C seemed necessary to me for the entirety of this article. Hence, the known information in the Introduction. I hope, however, that the modified, as suggested, content of the Introduction changed the version of this part in line with the Reviewer's expectations.

Point 2. It is necessary to mention some supplements which could have a role in the nutritional supplementation for COVID-19. For example it is demonstrate that lactoferrin is a role in the prevention and management of viral infection. A good reference is “Sinopoli, A, Isonne, C, Santoro, MM, Baccolini, V. The effects of orally administered lactoferrin in the prevention and management of viral infections: a systematic review. Rev Med Virol. 2022; 32( 1):e2261. https://doi.org/10.1002/rmv.2261”.

Response 2. The text has been supplemented as suggested by the Reviewer – vide p. 12.

Point 3. Also, Vitamin D should be mentioned in the possible role of SARS Cov 2. Good references are “Vimaleswaran, K. S., Forouhi, N. G., & Khunti, K. (2021). Vitamin D and covid-19. Bmj, 372”, “Wise, J. (2020). Covid-19: Evidence is lacking to support vitamin D’s role in treatment and prevention.

Response 3. The text has been supplemented as suggested by the Reviewer – vide p. 12.